# Small but Nice–Seed Dispersal by Tamarins Compared to Large Neotropical Primates

**Eckhard W. Heymann** [1,*], **Lisieux Fuzessy** [2] and **Laurence Culot** [3]

1 Verhaltensökologie & Soziobiologie, Deutsches Primatenzentrum–Leibniz-Institut für Primatenforschung, 37077 Göttingen, Germany

2 CREAF—Centre de Recerca Ecològica i Aplicacions Forestals, Universitat Autònoma de Barcelona, 08193 Catalunya, Spain

3 Laboratório de Primatologia, Departamento de Biodiversidade, Universidade Estadual Paulista (UNESP), Rio Claro 13506-900, SP, Brazil

* Correspondence: eheyman@gwdg.de

**Abstract:** Tamarins, small Neotropical primates of the genera *Saguinus* and *Leontocebus*, have a mainly frugivorous-faunivorous diet. While consuming the pulp of a high diversity of fruit species, they also swallow seeds and void them intact, thus acting as seed dispersers. Here we compare different aspects of the seed dispersal ecology of tamarins with that of large Neotropical primates from the genera *Ateles* (spider monkeys) and *Lagothrix* (woolly monkeys). Due to their small body size, tamarins disperse seeds of a smaller size range, fewer seeds per defecation, and seeds from a smaller number of different plant species per defecation compared to these atelines. We discuss whether tamarin seed dispersal is redundant or complementary to seed dispersal by atelines. On the level of plant species, our comparisons suggest that redundancy or complementarity depends on the plant species concerned. On the habitat level, seed dispersal by tamarins and large New World primates is probably complementary. Particularly, since tamarins are capable of persisting in disturbed forests and near human settlements, they are more likely to contribute to the natural regeneration of such areas than larger primates.

**Keywords:** Callitrichidae; Atelinae; seed dispersal effectiveness; seed dispersal distance; germination; forest regeneration; secondary forest





## 1. Introduction

Seed dispersal is a crucial process for plant reproductive success, plant population dynamics and spatial genetic structure, natural regeneration of vegetation, and maintenance of biodiversity [1–3]. Plants have evolved several different mechanisms of seed dispersal [4], but in tropical forests, zoochory, particularly endozoochory, is the prevailing dispersal mode for trees and woody lianas [5–7]. Besides birds and bats, primates are a major group of seed dispersal vectors [5] due to the fact that all primates (except tarsiers) include fruit in their diet, either as a primary food source or as a complement to a folivorous, faunivorous or exudativorous diet [8].

The contribution that a specific dispersal vector makes to the future reproduction of a plant has been conceptualized as seed dispersal effectiveness, SDE [9,10]. SDE comprises a number of determinants that can be subsumed in its quantitative and qualitative components [9,10]. This includes variables such as body size, the degree of frugivory of the vector, and the effect of gut passage on seed viability (germination rates). Neotropical primates (Platyrrhini), all of which include fruits in their diet, albeit to variable degrees, are the exclusive or principal dispersal vectors for a large number of woody plant species [11]. They vary in body size from the tiny pygmy marmoset, *Cebuella pygmaea* (110 g), to the large atelines, some of which may exceed 10 kg [12,13].

In this paper, we compare seed dispersal by small Neotropical primates, namely by the tamarins *Leontocebus nigrifrons* (previously *Saguinus fuscicollis nigrifrons*; see [14] for change of taxonomy) and *Saguinus mystax* (Figure 1) with that of large Neotropical primates, namely spider monkeys, *Ateles*, and woolly monkeys, *Lagothrix*. Tamarins, spiders, and woolly monkeys are all highly frugivorous, with fruit pulp representing the major portion of their respective diets [12,15–18]. They overlap strongly in the taxonomic spectrum of plants they exploit for fruit consumption (Supplementary Materials), making it likely that there is also considerable overlap in the spectrum of plant species whose seeds they disperse. Therefore, we make the following specific comparisons: (a) Diversity of consumed and dispersed plant species; (b) Consistency of seed dispersal (i.e., frequency of presence of seeds in defecations); (c) Number of seeds of the same and of different plant species included in defecations; (d) Seed size; (e) Effect of gut passage on germination success and time; (f) Seed rain, i.e., the number of seeds dispersed per area and time; (g) Dispersal distances; (h) Dispersal sites and directed dispersal; (i) Dispersal into secondary forest; (j) Effects on plant spatial genetic structure.

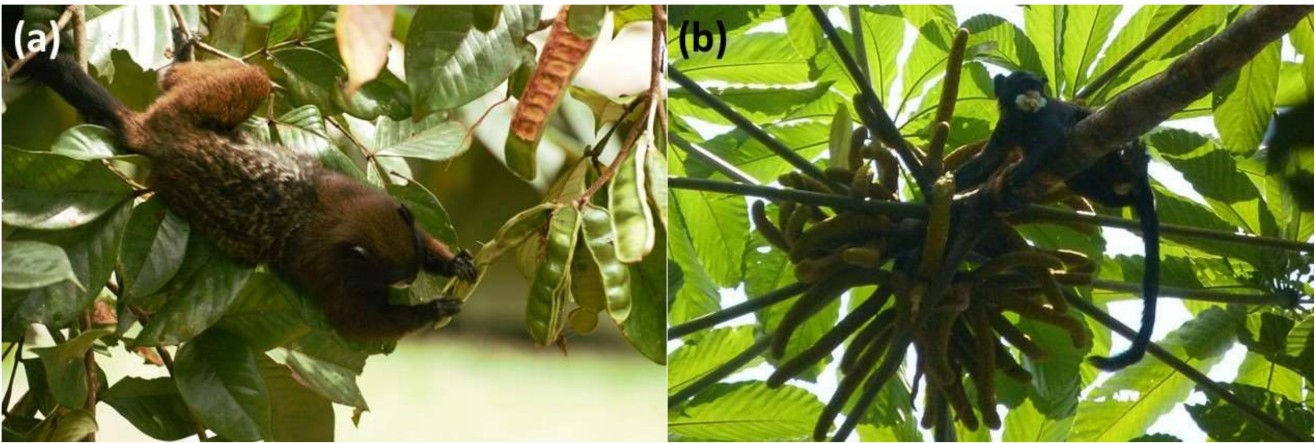

**Figure 1.** (**a**) *Leontocebus nigrifrons* feeding on *Inga* sp. (Fabaceae). Photo: Marco Tschapka. (**b**) *Saguinus mystax* feeding on *Cecropia sciadophylla* (Urticaceae). Photo: Eckhard W. Heymann.

## 2. Materials and Methods

This paper is based on our own research on the seed dispersal ecology of *L. nigrifrons* and *S. mystax*. Data for *Lagothrix* and Amazonian species of *Ateles* were extracted from pertinent references (provided in the tables), searched for in Google Scholar (search terms: "seed dispersal" + "genus or species name"), and browsed primatological and ecological journals. When available, we also consulted these. Altogether, we encountered a few references with data that could be used for the comparisons outlined in the Introduction. When available from the literature, we included data on seed dispersal by other tamarin species. We compiled a list of fruit plant species shared by *Leontocebus*, *Saguinus*, *Ateles*, and *Lagothrix* from references that provide lists of food species (see Appendix **??** for references). For this, we included data from *Leontocebus fuscicollis avilapiresi*, *Leontocebus weddelli*, *Saguinus imperator*, *Saguinus labiatus*, *Ateles belzebuth*, *Ateles paniscus*, *Lagothrix l. cana*, *Lagothrix l. lagothricha*, *Lagothrix l. lugens* and *Lagothrix l. poeppigii*. We restricted the lists of fruit plant species to primates distributed in Amazonia as to not confound comparisons with biogeographic differences.

## 3. Results and Discussion

### 3.1. Diversity of Consumed and Dispersed Fruit Species

The number of fruit species consumed and dispersed varies between species and between studies of the same species (Table 1). There is no obvious difference between the number of plant species exploited by tamarins and by atelines. However, the proportion

of plant species whose seeds are dispersed is lower in tamarins (52–65%) compared to atelines (83–99%). A potential problem of such a comparison is the length and temporal distribution of study periods. Longer studies are more likely to record a broader spectrum of food plant species and are more likely to capture plant species that are exploited only opportunistically or less intensively. In our own studies on *S. mystax* and *L. nigrifrons*, the number of plant species recorded in the diet and dispersed was around two times higher in the study by Culot [19]-a total of 24 months, each month represented twice compared to the study by Knogge [20] over 15 months. The lower proportion of plant species whose seeds are dispersed by tamarins is likely to be a consequence of the fact that they cannot swallow seeds beyond a certain size which, in contrast, can be swallowed by atelines (see Section 3.4 for the comparison of seed size).

*3.2. Consistency of Seed Dispersal*

Tamarins, spiders, and woolly monkeys almost continuously disperse seeds: 95–96% of tamarin defecations and 96–100% of defecations of *Ateles* and *Lagothrix* include at least one seed (Figure 2).

**Table 1.** Number of plant species exploited for fruit pulp, and number and percentage of plant species whose seeds are dispersed.

| Primate Species | Number of Plant Species Exploited for Pulp | Number of Dispersed Plant Species | % Dispersed | Study Periods and Duration [c] | Source |
|---|---|---|---|---|---|
| *Ateles belzebuth* | 90 | | 83 | 1992, duration not available | [21] |
| *Ateles belzebuth* | 73 | 72 | 99 | Aug 1995–Jul 1996, 457.45 h | [16] |
| *Ateles paniscus* | 151 | 138 | 91 | Apr 1977–May 1978, 135 d, 865 h | [22] |
| *Lagothrix l. lagothricha* | 183 | 165 | 90 [a] | Jan 1985–Sep 1987, 2400 h | [23] |
| *Lagothrix l. lugens* | 90 | | 84 | 1996–97, 720 h | [21] |
| *Lagothrix l. poeppigii* | 104 | 88 | 95 | Aug 1995–Jul 1996, 429,45 h | [16] |
| *Leontocebus nigrifrons*, 1994–95 | 124 | 81 | 65 | Mar 1994–May 1995, 74 d | [20] |
| *Leontocebus nigrifrons*, 2005–08 | 251 | 154 | 61 | Sep 2005–Feb 2006, Jun–Nov 2006, March–Aug 2007, Dec 2007–May 2008, 2303 h | [24] |
| *Saguinus mystax*, 1994–95 | 130 | 67 | 52 | Mar 1994–May 1995, 67 d | [16] |
| *Saguinus mystax*, 2005–08 | 267 | 151 | 57 | Sep 2005–Feb 2006, Jun–Nov 2006, March–Aug 2007, Dec 2007–May 2008, 2303 h | [24] |
| *L. nigrifrons* + *S. mystax* combined, 1994–95 [b] | 155 | 88 | 57 | | [16] |
| *L. nigrifrons* + *S. mystax* combined, 2005–08 [b] | 307 | 166 | 54 | | [24] |

[a] % dispersed calculated from Table 4 in the source. [b] the two tamarin species live in stable mixed-species groups that share a home range. [c] as provided in the source.

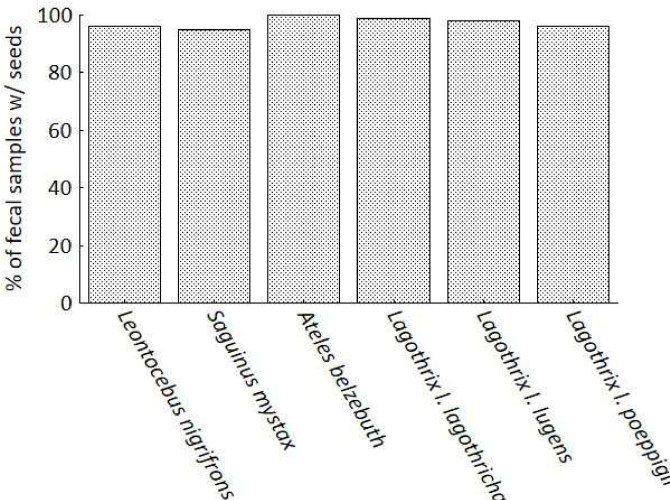

**Figure 2.** Percentage of fecal samples that include at least one seed. Data sources: *Leontocebus nigrifrons* and *Saguinus mystax*: [25]; *Ateles belzebuth*: [26]; *Lagothrix l. lagothricha*: [27]; *Lagothrix l. lugens*: [28]; *Lagothrix l. poeppigii*: [29].

*3.3. Number of Seeds and Seed Species Dispersed per Defecation*

Tamarins disperse a smaller number of seeds per defecation than spider and woolly monkeys. While the comparison is slightly hampered by the fact that studies variably report medians or means, in both cases, values are higher in the *Ateles* and *Lagothrix* (Table 2). The same is true for the number of seed species per defecation, but the difference is less pronounced (Table 3). Given the right-skewed distribution of the number of seeds or seed species per defecation, future studies should preferably report the median or both median and mean.

**Table 2.** Number of seeds per defecation.

| Primate Species | Parameter | Number of Seeds/ Defecation | Range | Sample Size (Number of Fecal Samples) | Source |
|---|---|---|---|---|---|
| *Ateles belzebuth* | median [a] mean [a] median [b] mean [b] | 7 39.1 5 7.8 | 0–402 | 738 | [26] |
| *Ateles chamek* | mean mean [b] | 249 22 | | 89 | [30] |
| *Lagothrix l. lagothricha* | Mean | 68 | 0– 3905 | 1397 | [27] |
| *Lagothrix l. lugens* | Mean | 79 | 0– 1409 | 1562 | [28] |
| *Leontocebus nigrifrons* | median | 1 | 1– 4800+ | 1137 | Knogge & Heymann, unpubl. data |
| | mean mean [b] | 2.2 1.7 | 1- 4800+ | 1137 | [25] |
| *Saguinus mystax* | median | 1 | 1– 4800+ | 924 | Knogge & Heymann, unpubl. data |
| | mean mean [b] | 2.2 1.5 | 1- 4800+ | 924 | [25] |
| *Saguinus leucopus* | | | 1–504 | 134 | [31] |

[a] seeds >1 mm; # seeds >3 mm. [b] without fecal samples including ≥10 seeds.

**Table 3.** Number of seed species per defecation.

| Primate Species | Parameter | Number of Seed Species/ Defecation | Range | Sample Size (Number of Fecal Samples) | Source |
|---|---|---|---|---|---|
| *Ateles belzebuth* | mean ± SD [a] | 1.9 ± 1.3 | 0–7 | 738 | [26] |
| *Ateles chamek* | mean | 2.2 | | 89 | [30] |
| *Lagothrix l. lagothricha* | mean ± SD median [b] | 1.8 ± 1.0 2 | 0–5 | 176 | [32] |
| *Lagothrix l. lagothricha* | mean ± SD | 2.4 ± 1.3 | 0–9 | 1397 | [27] |
| *Lagothrix l. lugens* | mean | 2.5 | 0–9 | 1562 | [28] |
| *Leontocebus nigrifrons* | median | 1 | | 1137 | Knogge & Heymann, unpubl. data |
| | mean | 1.2 | | 1137 | [25] |
| *Saguinus mystax* | median | 1 | | 924 | Knogge & Heymann, unpubl. data |
| | mean | 1.2 | | 924 | [25] |

[a] seeds >1 mm. [b] calculated from Figure 1 in the source.

### 3.4. Seed Size

Tamarins swallow seeds between <1 mm in diameter and >1 cm wide and >2 cm long (Figure 3), while atelines swallow seeds of almost up to twice the width and length [33] (Table 4). When considering that tamarins are roughly fifteen to twenty times smaller than a spider and woolly monkeys, they disperse very large seeds for their size. Even if a quite large overlap in the size of dispersed seeds exists, tamarins certainly cannot substitute the role of large atelines in dispersing the largest seeds available in the Amazonian Forest. To better understand the relative role of tamarins and large atelines in dispersing seeds of different sizes, future studies should compare the proportion of seeds of different sizes dispersed by primate species sharing the same habitat. Indeed, plant species composition directly influences the size of seeds available in the habitat and can thus hamper such comparisons. Secondly, a species that rarely disperse large seeds does not have the same functional effect as another that constantly disperses them.

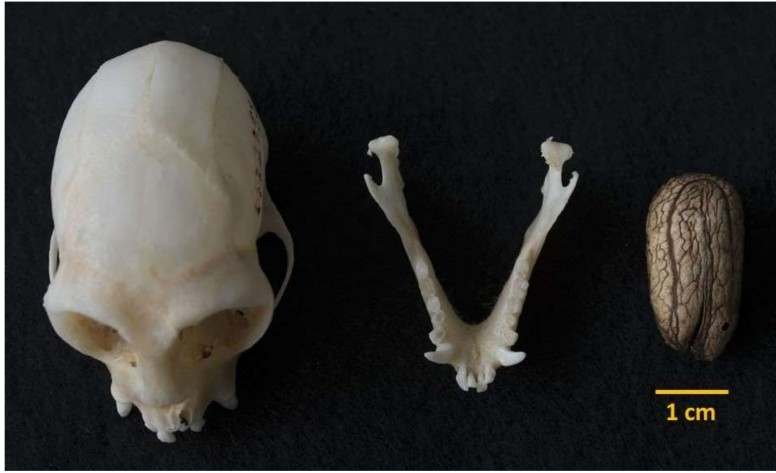

**Figure 3.** Skull and mandible of a *Leontocebus illigeri* (which is identical in size to *Leontocebus nigrifrons*; see measurements in [34] and a seed of *Abuta* sp., recovered from the defecation of a *L. nigrifrons*. Foto by Karin Tilch, © Deutsches Primatenzentrum–Leibniz-Institut für Primatenforschung.

**Table 4.** Size of dispersed seeds.

| | Dimension | Parameter | Size [cm] | Range | Sample Size [Number of Species] | Source |
|---|---|---|---|---|---|---|
| *Ateles belzebuth* | length | | | <0.1–3.9 | 133 | [26] |
| | length | | | <0.1–3.2 | 71 | [21] |
| *Ateles paniscus* | length<br>width | mean<br>mean | 1.38<br>1.02 | 0.01–2.76<br>0.01–2.00 | 12 | [35] |
| *Lagothrix l. lagothricha* | length<br>width | | | 0.8–2.13<br>0.62–1.48 | 11 | [32] |
| *Lagothrix l. lagothricha* | length | | | <0.1–4.00 | 80 | [21] |
| *Lagothrix l. lagothricha* | length<br>width | median<br>median | 1.10 | <0.1–3.30<br><0.1–1.80 | n.a.<br>n.a. | [36] |
| *Lagothrix l. lugens* | width | | | 0.07–2 | 28 | [37] |
| *Leontocebus nigrifrons* | lengthwidth | median<br>median | 1.38<br><br>0.80 | | 75 | Knogge & Heymann, unpubl. data |
| | length<br>width | mean<br>mean | 1.24 ± 0.56<br>0.75 ± 0.31 | 0.06–2.33<br>0.06–1.35 | 75 | [25] |
| *Saguinus mystax* | length<br>width | median<br>median | 1.380.80 | | 65 | Knogge & Heymann, unpubl. data |
| | length<br>width | mean<br>mean | 1.25 ± 0.59<br>0.74 ± 0.32 | 0.06–2.35<br>0.06–1.31 | 65 | [25] |
| | length<br>width | mean<br>mean | 1.19 ± 0.23<br>0.65 ± 0.10 | 0.73–1.95<br>0.45–0.97 | n.a. | [38] |
| *Saguinus geoffroyi* | length<br>width | mean<br>mean | 1.12 ± 0.41<br>0.62 ± 0.20 | 0.26–2.3<br>0.24–1.2 | n.a. | [38] |
| *Saguinus leucopus* | length?[b] | mean | 1.1 | <0.1–2.6 | 44 | [31] |
| *Saguinus niger* | length<br>width | | | 0.5–1.7 [a]<br>0.2–1.1 [a] | 6 | [39] |

[a] Range of means from [38]; since there are SDs around these means, the actual range is slightly larger. [b] Dimension not quoted in source.

*3.5. Germination Success and Time*

For most of the seed species tested, the passage through the tamarin gut has a neutral effect on germination success and time [39]. Seed passage through ateline guts usually increases or has no effect on germination success and time (Table 5). These results are in line with [40], who showed that frugivore primates (such as large ateline monkeys) increase seed germination by 75% after gut passage, while frugivore-insectivore primates (such as tamarins) have a neutral effect. The same study indicated that frugivore and frugivore-insectivore primates do not affect germination time. Overall, large atelines tend to have a more positive effect on plant reproduction than tamarins, but the latest usually do not have a negative effect either.

**Table 5.** Gut effects on seed germination in terms of success (proportion of germinated seeds) and on mean germination time (in days).

| Primate Species | Measure | Total Number of Plant Species | Number of Species Enhanced [a] | Number of Species Reduced [b] | Number of Neutral Effects | Source |
|---|---|---|---|---|---|---|
| *Ateles belzebuth* | Success<br>Time | 15<br>12 | 6<br>1 | 4<br>2 | 5<br>9 | [21] [c] |
| | Success<br>Time | 10<br>10 | 4<br>8 | 0<br>2 | 6<br>0 | [41] [c] |
| *Lagothrix l. lagothricha* | Success<br>Time | 16<br>9 | 8<br>1 | 1<br>3 | 7<br>5 | [21] [c] |
| *Lagothrix l. lugens* | Success | 21 | 8 | 1 | 12 | [42] [d] |
| *Leontocebus nigrifrons* | Success<br>Time | 27<br>27 | 5<br>3 | 6<br>2 | 16<br>22 | [43] [c,e] |
| *Saguinus mystax* | Success<br>Time | 24<br>24 | 3<br>7 | 3<br>0 | 18<br>17 | [43] [c,e] |

[a] increase in the proportion of germinated seeds and/or in the mean germination time for each species tested. [b] decrease in the proportion of germinated seeds and/or in the mean germination time for each species tested. [c] Significance reported by authors. [d] Significance calculated by chi-square using germination proportions reported by the author. [e] Germination time in weeks. Authors considered the effects of gut passage on germination time neutral if differences between treatment and control were less than four weeks. In these cases, no statistics were run to test the difference.

*3.6. Seed Rain*

The seed rain created by *L. nigrifrons* and *S. mystax* is intermediate between that created by *A. belzebuth* and *L. lagothricha* (Table 6). Notably, despite similar body size, seed rains created by *A. belzebuth* and *L. lagothricha* differ strongly, perhaps as a consequence of different population densities [29]. Stevenson [28] used a correction factor to include nighttime defecations and incomplete sampling of feces. Applying a similar correction to other studies would increase the number of dispersed seeds but not principally change the overall differences between species. In any case, these numbers are likely to have a large error margin, and they will obviously vary between populations and species as a function of population density. The seed rain is generally determined overall plant species dispersed. However, to be ecologically more meaningful, the seed rain should be determined per plant species. This would allow for a more specific comparison of the effect of the different primate species on plant populations.

**Table 6.** Seed rain expressed as the number of seeds dispersed per ha and year. All values rounded to the nearest 100.

| Primate Species | Number of Seeds Dispersed /Year/ha [a] | Source |
|---|---|---|
| *Ateles belzebuth* | 13,500 | [26] |
| | 2400 [b] | [29] |
| *Lagothrix l. lagothricha* | 255,900 [c]<br>10,700 [d] | [27] |
| *Lagothrix l. lugens* | 91,250 [e] | [42] |
| | 3,884,700 [f] | [28] |
| *Lagothrix l. poeppigii* | 3200 [b] | [29] |
| *Leontocebus nigrifrons* | 21,900 | [20] |
| *Saguinus mystax* | 27,100 | [20] |
| *L. nigrifrons + S. mystax* combined [g] | 49,000 | [20] |

[a] all studies included one group per species. Since in all species, home ranges overlap to variable degrees, the seed rain may be higher in overlap areas. [b] only seeds >5 mm in length. [c] calculated from 70,100 seeds/$km^2$/day mentioned in source. [d] calculated from 2940 seeds/$km^2$/day for seeds >3 mm. [e] calculated from 25,000 seeds/$km^2$/day mentioned in source. [f] calculated from 1,064,289 seeds/$km^2$/day mentioned in source. [g] the two tamarin species live in stable mixed-species groups that share a home range.

### 3.7. Seed Dispersal Distances

Seed dispersal distances, both mean or median and maximum, are approximately 2–3 times larger in spider and woolly monkeys compared to tamarins (Table 7). The seed dispersal distances determined in two independent studies on tamarins are highly consistent (173 m vs. 183 m and 152 m vs. 160 m), despite discrepant sample sizes. This suggests that the differences between tamarins and atelines are not due to the comparatively smaller sample sizes in the latter.

**Table 7.** Seed dispersal distances.

| Primate Species | Parameter | Dispersal Distance [m] | Maximal Dispersal Distance [m] | Sample Size (Number of Dispersal Events) | Source |
|---|---|---|---|---|---|
| *Ateles belzebuth* | mean ± SD<br>median | 443 ± 334<br>318 | 1281 | 186 | [26] |
| *Ateles paniscus* [a] | mean ± SD | 252 ± 145 | | 116 | [44] |
| *Lagothrix l. lagothricha* | mean [b] | 377 | 1106 | 98 | [32] |
| *Lagothrix l. lagothricha* | mean | 577 | 1540 | 66 | [27] |
| *Lagothrix l. lugens* | mean ± SD | 354 ± 213 | 989 (1466 [c]) | 264 | [42] |
| *Leontocebus nigrifrons,* 1994–95 | median | 173 | 638 | 1059 | [45] |
| *Leontocebus nigrifrons,* 2005–08 | median | 183 | 593 | 349 | [45] |
| *Saguinus mystax,* 1994–95 | median | 152 | 709 | 817 | [45] |
| *Saguinus mystax,* 2005–08 | median | 160 | 585 | 301 | [45] |
| *Leontocebus nigrifrons/ Saguinus mystax* | | | 513 | 39 | [46] |
| *Saguinus leucopus* | mean ± SD | 206 ± 95 | 529 | 51 | [31] |

[a] only seeds of *Ziziphus Cinnamomum*. [b] weighted mean of means for five plant species provided in Table 6 in [32]. [c] for abortive seeds of *Pithecellobium saman*.

### 3.8. Seed Dispersal Sites

The iterative use of certain locations or areas within the home range, e.g., resting and sleeping sites, is likely to affect spatial patterns of seed dispersal. The density of seeds and seedlings is higher in tamarin resting areas compared to other areas [47]. The density of seedlings and saplings of *Parkia panurensis* (Fabaceae) was significantly higher below sleeping sites of *L. nigrifrons* and *S. mystax* compared to control sites but lower compared to parent trees [48]. The number of saplings >1 m in height was almost higher below sleeping sites compared to parent trees (17 vs. 10). The repeated use of the same sleeping trees by *A. belzebuth* generates high densities of seeds of *Oenocarpus bataua* (Arecaceae), a preferred food source, at these sites [49] (see also *3.11.*).

### 3.9. Directed Dispersal

We use the term directed dispersal in the narrow sense, i.e., dispersal to specific sites or microhabitats obligatory for germination and recruitment [50], like, e.g., the dispersal of mistletoe seeds to branches. The only case of directed dispersal has been reported for seeds of *Asplundia peruviana* (Cyclanthaceae) dispersed by *L. nigrifrons* [51]. This hemi-epiphyte grows on trunks; seeds deposited on the soil may germinate but are unlikely to result in recruitment. Since *L. nigrifrons* often use vertical substrates for locomotion [52,53] and may defecate while clinging to a trunk, they are more likely to disperse seeds to trunks than *S. mystax*. The light diarrhea caused by the pulp of *A. peruviana* helps seeds to stick to the trunk rather than drop to the ground. Since *Ateles* and *Lagothrix* are mainly traveling and foraging in the upper strata of the forest and rarely, if ever, cling to trunks [54–56], they are highly unlikely to disperse seeds to trunks. Directed dispersal is, however, conceivable in the case of mistletoes, and *Phoradendron* (Loranthaceae), which are consumed by *Ateles* [15], where rubbing the anus to get rid of the sticky seeds may result in the deposition on branches. Pertinent observations are lacking, and the very low importance of mistletoes in the diet makes it unlikely that dispersal by *Ateles* is important for mistletoes at all.

### 3.10. Dispersal into Secondary Forest

Tamarins, like most callitrichids, can tolerate high levels of forest disturbance; they use or may even prefer secondary forests, and they can persist close to human settlements [57]. Moreover, given their small body size, they are very rarely hunted. In contrast, *Ateles* and *Lagothrix* are strongly associated with tall mature forests and use disturbed or low forests very rarely, if at all [22,58,59]. Furthermore, they are strongly hunted for meat throughout the Neotropics [60]. Therefore, tamarins can make contributions to forest regeneration and are more likely to disperse seeds into the secondary forest than a spider and woolly monkeys. At our study site, *L. nigrifrons* and *S. mystax* started to use a regenerating buffalo pasture ca. 10 years after clear-felling of the area [61] and dispersed the seeds of 63 plant species into this secondary forest, including large seeds from plants involved in later stages of regeneration [24]. Nineteen percent of seeds dispersed by *Saguinus niger* are moved from the primary into the secondary forest [39], and *Saguinus leucopus* may disperse seeds into pastures [31]. Through genotyping of seedlings, juveniles, and adults of *Parkia panurensis*, Heymann and co-workers [61] showed that seed dispersal by *L. nigrifrons* and *S. mystax* (the only dispersers of *P. panurensis* at their site) into secondary forest actually results in regeneration.

### 3.11. Effects on Plant Population Genetics

Very few studies examined the effect of Neotropical primate seed dispersal on plant population genetics. Seed dispersal of *P. panurensis* by *L. nigrifrons* and *S. mystax* (the only dispersers of this plant species at the study site, the Estación Biológica Quebrada Blanco) creates a significant spatial genetic structure up to 300 m on the seedling/sapling stage and up to 100 m on the adult stage [62]. Seed pools of *O. bataua* resulting from dispersal by *A. belzebuth* to sleeping trees show a significant genetic structuring [49]; whether this translates into a significant SGS in later stages was not examined. Genetic differentiation

between subpopulations of seedlings of *Inga ingoides* (Fabaceae) was smaller in an area where *Ateles chamek* (in the source wrongly named *Ateles paniscus*) was present compared to an area where this disperser was absent (extinguished by hunting) [63].

## 4. Concluding Discussion

Tamarins, spider, and woolly monkeys strongly overlap in the fruit part of their diet. They are, therefore, also likely to overlap in their seed dispersal activity but also to show differences. The two major differences are the size of dispersed seeds and the seed rain. Given their larger body size, spider and woolly monkeys can swallow larger seeds and thus will disperse seeds of plant species that the tamarins might exploit but for which they cannot swallow and disperse the seeds. On the other hand, it is conceivable that the tamarins disperse seeds of understory trees and treelets that are generally unattractive as a food source for larger Neotropical primates or that may not be used due to a perceived or actual higher predation risk for atelines closer to the ground. Thus, on the habitat scale, seed dispersal by tamarins vs. spider and woolly monkeys can be both complementary and redundant. On this scale, the importance of tamarins may primarily rest in their capability of using degraded areas and secondary forests in the early stages of regeneration and dispersing seeds from the primary into the secondary forest [24]. This can contribute to the natural regeneration process, which in a later successional stage will also allow larger Neotropical primates to use such a forest.

Spider and woolly monkeys produce a much more intensive seed rain than tamarins, also a consequence of their larger body size. Again, it is conceivable that the tamarins produce a seed rain for a number of plant species from the understory not or rarely dispersed by larger Neotropical primates, thus acting complementary to the spider and woolly monkeys.

Ideally, one would compare the SDE of tamarins and larger Neotropical primates, on the plant species level, at the same study site. This would reduce the number of confounding factors, e.g., differences between plant populations and individuals of the same species, soil properties, etc. In the absence of studies that allow such comparisons, our evaluation must remain tentative. We suggest that the SDE is similar between tamarins and larger Neotropical primates but that the different quantitative and qualitative components vary. While tamarins disperse obviously fewer seeds than larger Neotropical primates, single defecations include very few seeds, meaning that the clumping of seeds is strongly reduced. Even though the larger defecations of atelines attract more dung beetles and thus rates of secondary dispersal may be higher [64,65], the small distances of secondary dispersal would still mean that at some point of time during the development of seedlings and saplings into larger plants, intra-, and interspecific competition can emerge, resulting in only one of the many seeds dispersed will actually result in the recruitment of a reproductive adult plant.

We hope that our comparison stimulates comparative research of tamarin and ateline seed dispersal or more sophisticated approaches where the seed dispersal in general and the SDE in particular of primate communities are compared in order to obtain a better understanding of the relative contributions of different primate species to the natural regeneration processes of tropical forests.

**Supplementary Materials:** The following supporting information can be downloaded at: https://www.mdpi.com/article/10.3390/d14121033/s1. This Supplementary Material lists the plant genera shared by either or both tamarin genera and either or both ateline genera. Only genera where fruit pulp is consumed are included. Genera are included independent of their dispersal mode. The assignment of genera to families in this table may differ from the assignment in the source. We updated family assignments through a search at http://www.worldfloraonline.org [accessed on 18 November 2023]. Plant genera shared by all four primate genera are in dark orange, those shared by three genera in a medium orange, and those shared by one tamarin and one ateline genus are in light orange.

**Author Contributions:** E.W.H. and L.C. conceived the study; E.W.H., L.C. and L.F. collected, compiled, and analyzed data and wrote the manuscript. All authors have read and agreed to the published version of the manuscript.

**Funding:** Field research by E.W.H. was funded by the Deutsche Forschungsgemeinschaft (grant numbers HE 1870/3-(1-3), HE 1870/15-1,2, HE 1870/19-1, HE 1870/20-1, HE 1870/27-1). Field research by L.C. was funded by FRIA (Fonds pour la formation à la recherche dans l'industrie et dans l'agriculture) and FNRS (Fonds National de la Recherche Scientifique), Belgium, LC receives a Research Productivity Fellowship from CNPq (#314964/2021-5). LF receives funding from the European Union's Horizon 2020 research and innovation program under the Marie Skłodowska-Curie grant agreement (#101030199).

**Acknowledgments:** We thank Renato Hilário and João Pedro Souza-Alves for their invitation to contribute to this special issue. We are grateful to our Peruvian field assistants Camilo Flores Amasifuén, Jeisen Shahuano Tello, and Ney Shahuano Tello for their skillful and tireless support in the field, particularly for collecting the large numbers of tamarin poo needed to determine various aspects of SDE. Finally, we thank four anonymous reviewers for their helpful comments on the manuscript.

**Conflicts of Interest:** The authors declare no conflict of interest.

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
