# Peer review of "Small but Nice–Seed Dispersal by Tamarins Compared to Large Neotropical Primates"

_diversity, doi:10.3390/d14121033_

Round 1
Reviewer 1 Report
I think that the study is well written, clear and it should help as an incentive for future studies. At the beginning I was reluctant that the non-detailed methodology could be a major limitation for publication. However, given the semiquantitative nature of the comparison (i.e. lacking statistical analyses) I see no major inconvenient for publication after minor changes and the inclusion of the criteria for the studies to be included (such as study duration or sampling size, search engines used, where dissertation thesis included?). Adding this information will help in the replicability of the study for future comparisons. In addition, some minor points should be considered:
Title
I think that the title should show somehow that small refers to tamarin seed dispersal (perhaps: Small but nice - seed dispersal by tamarins in comparison with large Neotropical primates)
Abstract
I think it is important to include in the abstract that tamarins cannot disperse large seeds as atelines do.
Introduction
Line 31. A recent study (Correa et al. 2022) may strengthen this point, as the authors categorize seed dispersal modes for more than 4000 Amazonian tree species.
Line 38. As body size and degree of frugivory are not directly assessed as a component of SDE, I suggest changing them for the probability of seedling and sapling survival in different environments (perhaps after seed viability). These factors (body size and degree of frugivory) may be introduced latter on (perhaps body size is already included).
Figure 1a. The plant species is not Inga edulis (it is an Inga sp., but without a voucher it is difficult to determine the species. Figure 1b. The plant species is Cecropia sciadophylla (one of the very few Cecropia species with compound palmate leaves), not C. obtusifolia.
Materials and Methods
Line 74. More information is required in the methods section. For example, an adequate comparison of taxonomic diversity needs to consider sample size (as the number of consumed and dispersed species increases with sampling effort). Thus, it is important to mention if sampling size was assessed or at least state that the comparison on the number of species consumed or dispersed included yearly study periods (cf. Gonzalez et al. 2016).
I think that it is necessary to include additional information on the kind of data used. For instance, 1. how seed size was treated from the beginning (e.g. mean, median, smallest and largest length and width?). 2. The minimum number of species assessed in germination trails for a study to be included in the analyses. 3. The minimum number dung samples for a study to be included in the analyses of the proportion of fecal samples with dispersed seeds.
Results
Line 104. It would be interesting to discuss why Garber (1986) found few seeds during his studies of seed dispersal by tamarins.
Line 252. Change comma for a period (after the parenthesis)
Line 256. Although not published, I have seen Ateles and Lagothrix using pastures after 10-12 y forest recovery in areas surrounded by close mature forests (not very different from the observations on tamarins!). This, reinforces the idea that their role in forests regeneration is not as persistent as in the case of tamarins, because atelines are usually extirpated in disturbed sites.
Appendix A. Please exclude the abioticaly dispersed plant species (e.g. Bignoniaceae). The current name of the families of some genera should be updated (e.g. Casearia -Salicaceae-, Siparuna -Siparunaceae-, and Quiina -Ochnaceae-) and acknowledge that some genera have changed substantially (e.g. Maytenus, Licania, Crepidospermum and perhaps Tetragastris) but I will use the nomenclature used in the original sources (as it is).
Additional references
Correa, D. F., Stevenson, P. R., Umaña, M. N., Coelho, L. D. S., Lima Filho, D. D. A., Salomão, R. P., ... & Malhi, Y. (2022). Geographic patterns of tree dispersal modes in Amazonia and their ecological correlates. Global Ecology and Biogeography.
Gonzalez, M., Clavijo, L., Betancur, J., & Stevenson, P. R. (2016). Fruits eaten by woolly monkeys (Lagothrix lagothricha) at local and regional scales. Primates, 57(2), 241-251.
Author Response
I think that the study is well written, clear and it should help as an incentive for future studies. At the beginning I was reluctant that the non-detailed methodology could be a major limitation for publication. However, given the semiquantitative nature of the comparison (i.e. lacking statistical analyses) I see no major inconvenient for publication after minor changes and the inclusion of the criteria for the studies to be included (such as study duration or sampling size, search engines used, where dissertation thesis included?). Adding this information will help in the replicability of the study for future comparisons. In addition, some minor points should be considered:
We are grateful to this reviewer for his/her very constructive comments.
Title
I think that the title should show somehow that small refers to tamarin seed dispersal (perhaps: Small but nice - seed dispersal by tamarins in comparison with large Neotropical primates)
Title changed to “Small but nice - seed dispersal by tamarins in comparison to large Neotropical primates”
Abstract
I think it is important to include in the abstract that tamarins cannot disperse large seeds as atelines do.
We added the following sentence: “Due to their small body size, tamarins disperse seeds of a smaller size range, fewer seeds per defecation, and seeds from a smaller number of different plant species per defecation compared to these atelines”
Introduction
Line 31. A recent study (Correa et al. 2022) may strengthen this point, as the authors categorize seed dispersal modes for more than 4000 Amazonian tree species.
We have added this reference.
Line 38. As body size and degree of frugivory are not directly assessed as a component of SDE, I suggest changing them for the probability of seedling and sapling survival in different environments (perhaps after seed viability). These factors (body size and degree of frugivory) may be introduced latter on (perhaps body size is already included).
Schupp et al. (2010) include body size as a variable in their hierarchical flow chart representing the determinants of SDE, directly influencing the number of seeds dispersed. This conceptualization by Schupp et al. is a major argument for doing the comparison of tamarins with large atelines. We would weaken our line of reasoning by leaving this out at this place. We agree that the probability of seedling and sapling survival is an important factor which ideally would be compared as well. But there are simply too few data to undertake such a comparison. Therefore, we prefer not to make the change suggested by the reviewer.
Figure 1a. The plant species is not Inga edulis (it is an Inga sp., but without a voucher it is difficult to determine the species. Figure 1b. The plant species is Cecropia sciadophylla (one of the very few Cecropia species with compound palmate leaves), not C. obtusifolia.
The reviewer is definitely correct with regard to Cecropia. We are not quite sure whether this is also true for Inga. Nevertheless, we have made both changes as suggested.
Materials and Methods
Line 74. More information is required in the methods section. For example, an adequate comparison of taxonomic diversity needs to consider sample size (as the number of consumed and dispersed species increases with sampling effort). Thus, it is important to mention if sampling size was assessed or at least state that the comparison on the number of species consumed or dispersed included yearly study periods (cf. Gonzalez et al. 2016).
We added “Study duration [# of months]” in Table 1.
I think that it is necessary to include additional information on the kind of data used. For instance, 1. how seed size was treated from the beginning (e.g. mean, median, smallest and largest length and width?). 2. The minimum number of species assessed in germination trails for a study to be included in the analyses. 3. The minimum number dung samples for a study to be included in the analyses of the proportion of fecal samples with dispersed seeds.
We are not quite sure what the reviewer refers to in his point 1. We extracted the information that was available in the references, be it mean, median and/or range.
We used all available information and did not impose a restriction (minimum number of species or minimum number of fecal samples). In Fig. 2, we did not use data by Garber (1986), as the number of fecal samples is not quoted. However, we added data on seed size from Garber (1986) to Table 4.
We also added information on our search strategies: “… searched for in Google Scholar (search terms: “seed dispersal” + “genus or species name”), and browsed primatological and ecological journals. When available we also consulted theses. Altogether, we encountered few references with data that could be used for the comparisons outlined in the Introduction.
Results
Line 104. It would be interesting to discuss why Garber (1986) found few seeds during his studies of seed dispersal by tamarins.
Line 104?? Anyway, the tamarins studied by Garber were much less habituated than the tamarins at our site. Therefore, we could collect much more fecal samples.
Line 252. Change comma for a period (after the parenthesis)
Done
Line 256. Although not published, I have seen Ateles and Lagothrix using pastures after 10-12 y forest recovery in areas surrounded by close mature forests (not very different from the observations on tamarins!). This, reinforces the idea that their role in forests regeneration is not as persistent as in the case of tamarins, because atelines are usually extirpated in disturbed sites.
Thank you for this information. We are not sure whether this could be built in as “personal communication from anonymous reviewer”, which would be quite unusual and perhaps unacceptable.
Appendix A. Please exclude the abioticaly dispersed plant species (e.g. Bignoniaceae). The current name of the families of some genera should be updated (e.g. Casearia -Salicaceae-, Siparuna -Siparunaceae-, and Quiina -Ochnaceae-) and acknowledge that some genera have changed substantially (e.g. Maytenus, Licania, Crepidospermum and perhaps Tetragastris) but I will use the nomenclature used in the original sources (as it is).
Since the idea of Appendix A is to show the broad overlap in the genera consumed by tamarins and atelines, independent of dispersal mode, we prefer to retain abiotically dispersed genera. For clarification, we added the following sentence to the title of the table: “Genera are included independent of their dispersal mode”.
We updated the families for all genera, using worldfloraonline.org. This is stated as “Assignment of genera to families in this table may differ from the assignment in the source.
Additional references
Correa, D. F., Stevenson, P. R., Umaña, M. N., Coelho, L. D. S., Lima Filho, D. D. A., Salomão, R. P., ... & Malhi, Y. (2022). Geographic patterns of tree dispersal modes in Amazonia and their ecological correlates. Global Ecology and Biogeography.
Gonzalez, M., Clavijo, L., Betancur, J., & Stevenson, P. R. (2016). Fruits eaten by woolly monkeys (Lagothrix lagothricha) at local and regional scales. Primates, 57(2), 241-251.
Thank you for these references which we both have added.
Reviewer 2 Report
This study focused on the seed dispersal ecology of tamarin, spider, and woolly monkeys in the Neotropical region based on direct research results of L. nigrifrons and S. mystax, and databases extracted for other species from references.
Unfortunately, the author failed to apply appropriate research methods to analyze and summarize the records listed in the seven tables. I don’t understand why they did not use analytic tools to provide some conclusive results, based on which the whole issue could be discussed. What is provided in the Results and Discussion is just an introductory description of the records, which is not the way in science.
One of the research targets is to clarify whether tamarin seed dispersal is redundant or complementary is a wrong concept. Such an issue can only be addressed when all the seed dispensers in an ecosystem, including primates, nonprimate mammals, birds, and invertebrates, are considered and analyzed simultaneously, not just focusing on a few monkey species.
Author Response
This study focused on the seed dispersal ecology of tamarin, spider, and woolly monkeys in the Neotropical region based on direct research results of L. nigrifrons and S. mystax, and databases extracted for other species from references.
Unfortunately, the author failed to apply appropriate research methods to analyze and summarize the records listed in the seven tables. I don’t understand why they did not use analytic tools to provide some conclusive results, based on which the whole issue could be discussed. What is provided in the Results and Discussion is just an introductory description of the records, which is not the way in science.
We guess that with “appropriate research methods” the reviewer refers to statistics and meta-analyses. However, given the scarcity of data, this would be premature. For tamarin seed dispersal, there is practically only our own dataset, and for the atelines data also stem from very few studies. Thus, with the current data availability, there is no point for sophisticated analyses. Rather, at this point we feel it is necessary to draw the attention to the issue and hope that this stimulates further research that allows the application of “appropriate research methods” in the future.
One of the research targets is to clarify whether tamarin seed dispersal is redundant or complementary is a wrong concept. Such an issue can only be addressed when all the seed dispensers in an ecosystem, including primates, nonprimate mammals, birds, and invertebrates, are considered and analyzed simultaneously, not just focusing on a few monkey species.
The reviewer is only partially correct. Yes, there are plant species (e.g., figs) whose seeds are dispersed by large communities that include birds, bats, primates and other animals. However, there are also plant species for which primates are the only or the major dispersers (see PhD thesis Fuzessy). E.g., Leonia cymosa, an understory tree, is dispersed at our study site only by tamarins, and at a site in Ecuador only by tamarins and squirrel monkeys. Thus, the question of redundancy or complementarity clearly remains on the agenda. Ideally, we would have included other primates, e.g., capuchin monkeys, but data are available only from studies outside Amazonia.
Reviewer 3 Report
L53-62: Exclude markers and describe your comparisons in a textual way.
L73-74: Please, include references here.
L74: "When available, we extracted ..."
I believe many of the papers included did not have all information described here.
L80-81: Please, include references here.
L81-85: How relevant are your results considering that most of these species do not overlap their distributions? For instance, is it relevant that Ateles belzebuth show a complementary diet with Leontocebus nigricollis since they do not occur sympatrically? I understand that you may not found the same data for A. chamek, but this should be stated clearly in the text.
L92-93: There are several possible explanations. Differences in the region (habitat) studied, physical limitation for the manipulation of fruits, and study period.
L110: on >> one
L273: compared
L283: I have already seen large species such as muriquis come to the understory to feed on small fruits (e.g. Psicotria). However, I did not see spider monkeys doing the same. Maybe it varies based on predator risk or availability but not attractivity.
L284: Again, I emphasize that you don't evaluate seed dispersal in a habitat scale but on a large (regional?) scale. Many of the species you mentioned in this review do not inhabit the same habitats (e.g. A. belzebuth or paniscus inhabit forests in the Guyana shield while tamarins live in the forests in southwestern Amazonia). Thus, discuss complementarity and redundance do not seems appropriate here. Please, avoid to discuss your results on a habitat scale (that looks like a more refined [local] scale), unless you prefer to discuss just the species that effectively inhabit the same habitats (such as Lagothrix l. cana and poeppigii, Ateles chamek and the tamarins), thus reducing the number of confounding factors as you mentioned below.
L300: See my previous commentary.
Author Response
L53-62: Exclude markers and describe your comparisons in a textual way.
Done
L73-74: Please, include references here.
All references are quoted in the tables. Therefore, we think that quoting them here would unnecessarily blow up the text.
L74: "When available, we extracted ..."
Done
I believe many of the papers included did not have all information described here.
That’s true. Therefore, the addition “When available, we extracted …” that you suggested is important.
L80-81: Please, include references here.
We added the following: “(see Appendix A for references)”
L81-85: How relevant are your results considering that most of these species do not overlap their distributions? For instance, is it relevant that Ateles belzebuth show a complementary diet with Leontocebus nigricollis since they do not occur sympatrically? I understand that you may not found the same data for A. chamek, but this should be stated clearly in the text.
Ateles belzebuth and Leontocebus nigricollis occur sympatrically in many areas of Western Amazonia, e.g. Tiputini (but unfortunately only seed dispersal by A. belzebuth has been studied there. At our site in Peru, Leontocebus nigrifrons and Saguinus mystax are also sympatric with L. lagothricha poeppigii and A. belzebuth, but the latter have not been studied here. As we show with Appendix A, the diets of Saguinus, Leontocebus and the atelines are strongly overlapping (not complementary). Since there is no reason to assume that seed dispersal by e.g., A. belzebuth is very different from A. chamek, or very different between species of Leontocebus and Saguinus, we think that in the absence of “sympatric” data, our comparisons still can reveal some general patterns or trends. Obviously, only parallel studies of tamarins and atelines at the same site will solve the issue. This is what we actually hope to stimulate with our paper.
L92-93: There are several possible explanations. Differences in the region (habitat) studied, physical limitation for the manipulation of fruits, and study period.
We added the following sentence at the end of this paragraph: “The lower proportion of plant species whose seeds are dispersed by tamarins is likely to be a consequence of the fact that they cannot swallow seeds beyond a certain size which in contrast can be swallowed by atelines (see 3.4 for the comparison of seed size)”.
L110: on >> one
Done
L273: compared
Done
L283: I have already seen large species such as muriquis come to the understory to feed on small fruits (e.g. Psicotria). However, I did not see spider monkeys doing the same. Maybe it varies based on predator risk or availability but not attractivity.
We have added the following statement: “… or that may not be used due to a perceived or actual higher predation risk for atelines closer to the ground”
L284: Again, I emphasize that you don't evaluate seed dispersal in a habitat scale but on a large (regional?) scale. Many of the species you mentioned in this review do not inhabit the same habitats (e.g. A. belzebuth or paniscus inhabit forests in the Guyana shield while tamarins live in the forests in southwestern Amazonia). Thus, discuss complementarity and redundance do not seems appropriate here. Please, avoid to discuss your results on a habitat scale (that looks like a more refined [local] scale), unless you prefer to discuss just the species that effectively inhabit the same habitats (such as Lagothrix l. cana and poeppigii, Ateles chamek and the tamarins), thus reducing the number of confounding factors as you mentioned below.
See response to comment on l. 81-85. Also, in the penultimate paragraph, we clearly classify our conclusions as tentative and demand that more definitive conclusions can only be reached by studies on sympatric tamarins and atelines.
L300: See my previous commentary.
See response to comment on l. 81-85
Reviewer 4 Report
Dear authors
I reviewed your manuscript entitle, “Small but nice – seed dispersal by tamarins and large Neotropical primates compared” and I recognize the contribution of this manuscript to our understanding on seed dispersal by Neotropical primates.
The manuscript is very well written and comprehensive. You provided very interesting information of seed dispersal and most important, the complementarity of seed dispersal by two sets of primate species.
There is very few editing to be done (on the body of the manuscript)
Regards

Author Response
Dear authors
I reviewed your manuscript entitle, “Small but nice – seed dispersal by tamarins and large Neotropical primates compared” and I recognize the contribution of this manuscript to our understanding on seed dispersal by Neotropical primates.
The manuscript is very well written and comprehensive. You provided very interesting information of seed dispersal and most important, the complementarity of seed dispersal by two sets of primate species.
Thank you very much for your appreciation!
There is very few editing to be done (on the body of the manuscript)
Regards
We have written our responses directly into the PDF file with the reviewer’s comments.
